# Characterization of the *mbsA* Gene Encoding a Putative APSES Transcription Factor in *Aspergillus fumigatus*

**DOI:** 10.3390/ijms22073777

**Published:** 2021-04-06

**Authors:** Yong-Ho Choi, Sang-Cheol Jun, Min-Woo Lee, Jae-Hyuk Yu, Kwang-Soo Shin

**Affiliations:** 1Department of Microbiology, Graduate School, Daejeon University, Daejeon 34520, Korea; youngho1107@gmail.com (Y.-H.C.); scjun@jbnu.ac.kr (S.-C.J.); 2Soonchunhyang Institute of Medi-Bio Science, Soonchunhyang University, Cheonan 31151, Korea; mwlee12@sch.ac.kr; 3Department of Bacteriology, University of Wisconsin-Madison, Madison, WI 53706, USA; 4Department of Systems Biotechnology, Konkuk University, Seoul 143-701, Korea

**Keywords:** APSES transcription factor, *Aspergillus fumigatus*, MbsA, rodlet layer, gliotoxin, virulence, transcriptomics

## Abstract

The APSES family proteins are transcription factors (TFs) with a basic helix-loop-helix domain, known to regulate growth, development, secondary metabolism, and other biological processes in *Aspergillus* species. In the genome of the human opportunistic pathogenic fungus *Aspergillus fumigatus*, five genes predicted to encode APSES TFs are present. Here, we report the characterization of one of these genes, called *mbsA* (Afu7g05620). The deletion (Δ) of *mbsA* resulted in significantly decreased hyphal growth and asexual sporulation (conidiation), and lowered mRNA levels of the key conidiation genes *abaA*, *brlA,* and *wetA*. Moreover, Δ*mbsA* resulted in reduced spore germination rates, elevated sensitivity toward Nikkomycin Z, and significantly lowered transcripts levels of genes associated with chitin synthesis. The *mbsA* deletion also resulted in significantly reduced levels of proteins and transcripts of genes associated with the SakA MAP kinase pathway. Importantly, the cell wall hydrophobicity and architecture of the Δ*mbsA* asexual spores (conidia) were altered, notably lacking the rodlet layer on the surface of the Δ*mbsA* conidium. Comparative transcriptomic analyses revealed that the Δ*mbsA* mutant showed higher mRNA levels of gliotoxin (GT) biosynthetic genes, which was corroborated by elevated levels of GT production in the mutant. While the Δ*mbsA* mutant produced higher amount of GT, Δ*mbsA* strains showed reduced virulence in the murine model, likely due to the defective spore integrity. In summary, the putative APSES TF MbsA plays a multiple role in governing growth, development, spore wall architecture, GT production, and virulence, which may be associated with the attenuated SakA signaling pathway.

## 1. Introduction

Members of APSES family (**A**sm1p, **P**hd1p, **S**ok2p, **E**fg1p, and **S**tuAp) transcription factors (TFs) regulate growth, morphogenesis, development, secondary metabolism, and other biological processes in fungi [1]. This group of proteins comprises a family of TFs that share a highly conserved helix-loop-helix DNA-binding motif [2,3,4,5]. Based on correlation of APSES domain, this proteins are divided into four major groups (clade A to D) [1]. Each of these proteins has been shown to play a critical role in governing fungal morphogenesis, development, and virulence.

In the genome of the opportunistic human pathogenic fungus *Aspergillus fumigatus*, five genes are predicted to encode APSES proteins: Afu7g05620, *rgdA*, *stuA*, *xbpA*, and *afpA* (Afu5g11390). The clade C-II APSES protein StuA controls asexual reproduction, conidiophore formation, and conidia germination [6]. Mutants deficient in *stuA* produce abnormal conidiophores and low numbers of dysmorphic conidia [6]. In addition, StuA regulates secondary metabolite biosynthetic genes, morphogenesis regulating genes, and allergen encoding genes [6,7]. RgdA, which belongs to clade A-I, also plays a multiple role in governing growth, development, fungal toxin production, and virulence via attenuation of PKA and SakA signaling [8]. The hyphal growth, asexual sporulation, and conidia germination are decreased by the absence of *rgdA*. Moreover, the conidia wall architecture and hydrophobicity are changed in the Δ*rgdA* mutant [8]. These findings suggest that in addition to functioning as a transcriptional repressor during conidiation, RgdA may govern other cellular processes in *A*. *fumigatus*. 

However, functions of the other APSES TFs including Afu7g05620 gene product in *Aspergillus* have remained to be elucidated. Previously, flucytosine-responsive Mbp1/Swi4-like protein was designated as Mbs1 in *Cryptococcus neoformans* [9]. So, we named Afu7g05620 (XP_748947.1) gene as *mbsA* (Mbp1- and Swi6-like protein A). In this study, we have characterized functions of the putative APSES gene *mbsA* in *A*. *fumigatus*. Our series of genetic, biochemical, genomic, histological, and virulence studies have revealed that MbsA plays a multiple role in governing cellular proliferation, asexual development, cell wall architecture, spore integrity, gliotoxin production, virulence, as well as proper SakA MAP kinase signaling.

## 2. Results

### 2.1. Summary of A. fumigatus MbsA

The ORF of *mbsA* in *A. fumigatus* Af293 (Afu7g05620) consists of 2806 bp nucleotides with two introns, and is predicted to encode an 898 amino acid length protein. As shown in Figure 1A, the predicted MbsA protein has a KilA-N domain (117 to 200 aa) in the N-terminal region, three ankyrin repeat-containing domains in the middle region (446 to 475, 516 to 559, 595 to 623 aa), and a coiled-coil domain at the C-terminal region. The KilA-N domain amino acid sequence of MbsA of *A. fumigatus* shows 97.6~100% identity with MbsA-like proteins of other *Aspergillus* species. On the other hand, it shows only 30.8~53.5% amino acid sequence identity with other APSES proteins of *A. fumigatus* (StuA and RgdA) and Mbp1 of *Saccharomyces cerevisiae* (data not shown). In the unrooted phylogenetic analysis based on the amino acid sequence of KilA-N domain, MbsA homologs of *Aspergillus* are clustered in the same group, RgdA of *A*. *fumigatus* (AfuRgdA) and Mbp1 of *S*. *cerevisiae* form another group, and StuA of *A*. *fumigatus* (AfuStuA) is distinctly related (Figure 1B). We examined levels of the *mbsA* mRNA throughout the lifecycle and found that there were no differences in all developmental phases except vegetative phase (Figure 1C).

### 2.2. MbsA Affects Vegetative Growth and Conidial Development

To investigate the functions of MbsA, the wild type (WT), *mbsA* deletion mutant, and complemented strains (C′) were inoculated onto solid glucose minimal medium with 0.1% yeast extract (MMY) and incubated for 3 days. The colony diameter of Δ*mbsA* mutant was smaller (about 25%) than that of WT and C′ strains (Figure 2A). The number of conidia per plate produced by Δ*mbsA* mutant was lower than that of WT and C′ strains (Figure 2B). In addition, mRNA expression levels of key asexual developmental regulators, *abaA*, *brlA*, and *wetA* were significantly lowered in the Δ*mbsA* mutant (Figure 2C). These results suggest that MbsA is required for proper vegetative growth and conidiation in *A*. *fumigatus*. We analyzed the germination rate of the Δ*mbsA* conidia in comparison to those of WT and C′ strains in triplicate. Germination rate of the Δ*mbsA* conidia was significantly reduced compared to other strains. Germination rate of the Δ*mbsA* conidia was 33% lower than that of WT and C′ strains. Moreover, while about 85% of WT and C′ strains’ conidia germinated, only 50% of the Δ*mbsA* spores germinated after 10 h incubation (Figure 2D).

### 2.3. MbsA Is Involved in Chtin Synthesis in A. fumigatus

To test a potential role of MbsA in chitin synthesis, we incubated WT, Δ*mbsA*, and C′ strains to solid yeast extract-glucose (YG) containing Nikkomycin Z (NZ). As shown in Figure 3A, the mutant strain showed a significantly reduced radial growth. To further dissect MbsA function in chitin synthesis, we analyzed mRNA levels of chitin synthesis-related genes. The mRNA levels of genes encoding chitin synthases were decreased significantly by loss of *mbsA* (Figure 3B). Taken together, these results suggest that MbsA is needed for the synthesis of chitin.

### 2.4. MbsA Positively Regulates the SakA MAP Kinase Pathway

To examine a possible relationship between MbsA and the SakA MAPK pathway that governs stress-response signaling, we analyzed phosphorylation levels of SakA and MpkA, and mRNA levels of SakA MAPK pathway related genes. We carried out anti-phospho-p38 and anti-phospho-p42/44 immunoblotting with total soluble protein extracts from WT, Δ*mbsA*, and C′ strains treated with 10 mg/mL Calcofluor White (CFW) for 20 min. A protein with SakA and MpkA predicted molecular mass became transiently phosphorylated in response to the cell wall stress. However, the phosphorylation levels of Δ*mbsA* strain was greatly lower than those of WT and C′ strains (Figure 4A). Transcript levels of SakA MAPK pathway–related genes examined were significantly decreased in the Δ*mbsA* mutant (Figure 4B). These results indicate that MbsA positively regulates the SakA MAPK pathway. 

### 2.5. MbsA Is Necessary for the Conidia Hydrophobicity and Plays a Role in rodA Expression

The capacity of conidia to reach alveoli is due to the hydrophobicity of conidia and the properties of hydrophobin affect conidial hydrophobicity. To understand the role of MbsA on governing conidial hydrophobicity, we analyzed hydrophobicity of WT, Δ*mbsA*, and C′ colonies using a detergent permeation assay and solvent partitioning assay. As shown in Figure 5A, the detergent solution began to permeate into the colony after 2 h and completely penetrated at 18 h in the Δ*mbsA* mutant. On the contrary, WT and C′ strains still made appeare the detergent solution droplets on the colony surface at 18 h. To further confirm this, we performed microbial adhesion to hydrocarbon (MATS) assay. Hydrophobicity of the Δ*mbsA* conidia was decreased about 16% compared to that of WT and C′ conidia (Figure 5B). Then, we extracted the conidial hydrophobin RodA from the conidia and analyzed by SDS-PAGE. The intensity of RodA band was greatly reduced in the Δ*mbsA* conidia indicating that the amount of the RodA protein was reduced by the loss of *mbsA* (Figure 5C). Levels of mRNA of the hydrophobin genes *rodA* and *rodC* were also significantly decreased in the Δ*mbsA* mutant (Figure 5D). These results suggest that MbsA positively regulates expression of the *rodA* and *rodC* genes and subsequently confers proper conidial hydrophobicity. 

### 2.6. MbsA Is Required for Proper Spore Wall Formation 

To characterize the link between the change of conidial hydrophobicity and conidia cell wall structure, we observed the fine structure of the conidia of the three strains by using transmission electron microscopy (TEM) and atomic force microscopy (AFM). Structure of the Δ*mbsA* conidia surface was strikingly different from that of WT and C′ conidia, as several irregular protrusions were observed on the surface of Δ*mbsA* conidia (Figure 6A). Conidial surfaces were further analyzed by AFM. In contrast to the WT and C′ conidia that are covered with a crystalline-like array of rodlets, the Δ*mbsA* mutant conidial surface was amorphous without any organized structure and the differences were more clearly visible in phase image (Figure 6B). These data suggest that MbsA is necessary for proper conidia wall architecture.

### 2.7. MbsA Down-Regulates Gliotoxin Biosynthesis

To further characterize the complex role of MbsA, we performed RNA-seq analysis using Δ*mbsA* and WT strains. Notably, transcripts levels of the gliotoxin (GT) biosynthetic genes *gliF* encoding a cytochrome P450 oxidoreductase and *gliA* encoding an MFS GT efflux transporter were over 100-fold higher in the Δ*mbsA* mutant than WT strain (Appendix A). As shown in Figure 7A, most of the GT biosynthetic clustered genes were up-regulated by Δ*mbsA*. To corroborate the RNA-seq results, we examined mRNA levels of four *gli* genes by RT-qPCR. Levels of *gliA*, *gliM*, *gliP*, and *gliT* transcripts were significantly higher (4 to 68-fold) in Δ*mbsA* strain compared to those of WT and C’ strains (Figure 7B) suggesting that the mutant strain may produce more GT than WT and complemented strain. To confirm this, we assessed levels of GT in WT, Δ*mbsA*, and C′ strains, and found that the Δ*mbsA* mutant produced about 5-fold more GT and other secondary metabolites compared to WT and C′ strains (Figure 7C). 

### 2.8. MbsA Plays an Important Role in Virulence

To investigate the pathological significance of the MbsA protein during *A*. *fumigatus* infection, conidia of WT, Δ*mbsA*, and C′ strains were intranasally introduced to neutropenic mice, which were generated by combinatorial administrations of cyclophosphamide and cortisone acetate (Figure 8A). Pathological outcomes were monitored by mouse survivability. In the survival curve analysis, while a group infected with WT and C′ strain showed the first mortality on 2.5 days after infection and displayed 10% survival rate within 3.5 days, a group infected with the Δ*mbsA* mutant showed the first death on 2.5 days and 90% survival rate even after 5 days (*p* = 0.0003) (Figure 8B). Next, to understand why the *mbsA* deletion led to less severity in mouse survivability, lung tissue sections were prepared from mice infected with three strains’ conidia and stained with Hematoxylin and Eosin (H&E) and Periodic acid-Schiff (PAS) to observe the extent of tissue damage and hyphal growth. In H&E staining, WT infected lung tissue showed severe lung damages with disruption of alveolar structure and bronchial wall and necrosis around bronchial region. However, compared to WT, Δ*mbsA* led to mild disruption of the basement of bronchial wall. C′ strain regained severe fungal damages which were similar to that caused by WT infection. Of note, as shown by PAS staining, the different lung tissue damages were associated with the extent of hyphal germination, which was much less in Δ*mbsA* strain infection than WT and C′ strain infection (Figure 8C). In addition, loss of *mbsA* significantly decreased (about 20-fold) the pulmonary fungal burden of mice (Figure 8D).

## 3. Discussion

The APSES TF family function as key regulators of differentiation, vegetative growth, and asexual and sexual development from plant to fungi. However, a systematic investigation of their roles in *A. fumigatus* is not fully performed yet. Among the five predicted APSES TFs, only StuA and RgdA have been studied in *A*. *fumigatus* [6,7,8,10,11,12]. MbsA of *A. fumigatus* is phylogenetically unrelated with RgdA of the same fungus, or Mbp1 of budding yeast *S*. *cerevisiae* (Figure 1). The deletion of *mbsA* resulted in decreased mycelial growth, conidiation, germination, and reduced mRNA levels of key asexual sporulation genes compared to WT (Figure 2). To elucidate the mechanism of delayed growth in Δ*mbsA* mutant, we analyzed the expression of *dpr* genes, which are down-regulated conidial germination [13]. The expression levels of three *dpr* genes were significantly increased by the loss of *mbsA*, suggesting that the mutant strain showed delayed growth may due to suppress conidial germination (Appendix A). Based on these, we propose that MbsA of *A*. *fumigatus* is necessary for normal growth and proper asexual development similar to RgdA.

Fungi are able to sense and respond external stress for survival in harsh environmental conditions and this ability is a key factor for the establishment of a successful infection [14]. The Δ*mbsA* strain was more sensitive to chitin synthesis inhibitor, Nikkomycin Z, and expressed lower levels of chitin synthesis genes (Figure 3). Chitin is a β-1,4-linked homopolymer of *N*-acetyl glucosamine (GlcNAc) and its synthesis is important for hyphal development. Chitin synthase (CHS) catalyzes GlcNAc polymerization from UDP-GlcNAc and CHS appears to play important roles in chitin synthesis in hyphal tips and conidia, as well as in polarized hyphal growth [15,16,17,18]. Deletion of CHS genes lead to reduced hyphal growth and periodic swellings along hyphal lengths [19].

Previous studies described that SakA plays an important role in fungal cell wall integrity. The deletion of *sakA* displayed alterations in cell wall composition and significant increased sensitivity to cell wall–damaging agents, and the cell wall–targeting antifungals caspofungin and nikkomycin Z [20]. In addition, activation of the MpkA cell wall integrity MAPK pathway in response to osmotic or cell wall stress was largely dependent on SakA and MpkC [20]. In *Aspergillus*, MpkA-RlmA signaling is involved in the transcriptional activation of cell wall–related genes, such as *chsA*, *chsC*, and *chsE* [21,22,23]. In accordance with these findings, the expression level of *rlmA* and the phosphorylation levels SakA and MpkA were lowered by the loss of *mbsA* (Figure 4). Collectively, we proposed that MbsA regulates cell wall integrity through properly activating the SakA MAPK pathway. 

Like RgdA, MbsA is also required for the conidia hydrophobicity, synthesis of the hydrophobin RodA protein, and conidia cell wall architecture (Figure 5 and Figure 6). The ability of air-borne conidia to reach alveoli is primarily dependent on the hydrophobic rodlets layer, which promotes the dispersion of spores [24]. In addition, the rodlet layer interferes with the recognition of spores by the human immune system [25]. In *A*. *fumigatus*, seven hydrophobins (RodA to RodG) are identified [26] and only RodA is responsible for the rodlet formation, permeability, hydrophobicity, and immune-inertia of conidia cell wall surface [26]. Deletion of *rodA* modifies the properties of the conidia cell wall surface and effects on the drug sensitivity of the fungi [26]. Despite *rodB* transcript being highly expressed in the absence of *mbsA*, the rodlet layer is not formed in the mutant strain indicating that RodB is not essential in the conidial hydrophobicity and formation of rodlets as previously reported [26]. Our results have revealed that MbsA regulates expression of RodA and formation of the proper rodlet layer, and the absence of *mbsA* may cause defects in fungal pathogenicity, and RgdA plays an identical role in these. 

The APSES TF StuA has been shown to positively regulate the clustered aflatoxin biosynthetic genes in *A*. *flavus* and several secondary metabolite biosynthetic cluster genes in *A*. *fumigatus* [7,27]. In contrast, MbsA in *A. fumigatus* represses expression of the GT clustered genes and production of GT, similar to RgdA [8] (Figure 7). As a result, the Δ*mbsA* mutant produced significantly higher amount of GT than WT and C′ strains. Interestingly, despite elevated GT production, the virulence of the mutant in the murine model was greatly reduced by the loss of *mbsA* (Figure 8). Our histological and fungal burden studies with the infected mice showed that the deletion of *mbsA* resulted in milder necrosis and disruption of bronchiole region compared to those found in WT and C′ strains. Furthermore, Δ*mbsA* significantly decreased the pulmonary fungal burden of mice (Figure 8). Collectively, regardless of the elevated levels of GT, the absence of *mbsA* leads to reduced virulence, which may be associated with the lowered conidial hydrophobicity and rodlet layer, and the attenuated SakA MAPK pathway. 

Taken together, MbsA plays similar roles with RgdA in governing fungal growth, differentiation, secondary metabolism, and virulence may be due to belong to same group (clade A) [28]. However, MbsA regulates the SakA MAP kinase pathway contrary to RgdA.

In summary, we propose a genetic model depicting the complex role of MbsA in *A*. *fumigatus* (Figure 9). In this model, MbsA positively acts at or upstream of the stress activated SakA MAPK signaling pathways, which activates chitin biosynthesis. While MbsA up-regulates asexual developmental activators and the subsequent RodA production, represses expression of the GT gene clusters and production of GT. Additional studies are needed to identify detailed molecular mechanism of MbsA and interaction with RgdA or other APSES TFs in *A*. *fumigatus*.

## 4. Materials and Methods

### 4.1. Strains and Culture Conditions

*A. fumigatus* AF293.1 (*AfpyrG1*) was used to generate the Δ*mbsA* mutant and AF293 was used as a wild type (WT). Fungal strains were grown on glucose minimal medium (MMG) or MMG with 0.1% yeast extract (MMY) with appropriate supplements as described previously [29]. For the production of a colony on solid medium, porous cellophane was deposited on the surface of the agar and point-inoculated with about 10^5^ conidia onto the cellophane. The petri dishes were incubated in the dark at 37 °C for 3 days. Fungal tissue was collected using a sterile spatula and used for RNA extraction. For asexual developmental induction, about 5 × 10^5^ conidia/mL of WT and relevant mutant strains were inoculated in 500 mL of liquid MMY and incubated at 37 °C and 250 rpm for 16 h. The mycelium was harvested by filtering through Miracloth (Calbiochem, San Diego, CA, USA), transferred to solid MMY, and incubated at 37 °C for air-exposed asexual developmental induction. Samples were collected at various time points post asexual developmental induction.

### 4.2. Generation of the ΔmbsA Mutant in A. fumigatus

The deletion construct generated employing double-joint PCR [30] containing the *A*. *nidulans* selective marker (*AnipyrG*) with the 5′ and 3′ franking regions of the *A*. *fumigatus mbsA* gene (Afu3g13920) was introduced into the recipient strains [31]. The selective marker was amplified from *A. nidulans* FGSC A4 genomic DNA with the primer pair oligo697/oligo698. The null mutant colonies were isolated and confirmed by diagnostic PCR (oligo378/oligo379), followed by restriction enzyme digestion. To complement the *mbsA* null mutant, a double-joint PCR (DJ-PCR) method was used [30] with *hygB* as the selective marker. The oligonucleotides used in this study are listed in Appendix A.

### 4.3. Nucleic Acid Isolation and Manipulation

Total RNA isolation and quantitative RT-PCR (RT-qPCR) assays were performed as previously described [32,33,34]. Briefly, each sample was homogenized in 1 mL of TRIzol reagent (Invitrogen, Waltham, MA, USA) using a Mini-Bead Beater (BioSpec Products, Bartlesville, OK, USA) and 0.3 mL of Zirconia/Silica beads (RPI, Mt. Prospect, IL, USA). The supernatant was mixed with an equal volume of iced isopropanol and centrifuged again. The RNA pellets were washed with 70% ethanol by diethyl pyrocarbonate (DEPC) treated water and dissolved in the RNase-free water. RNA quality was checked by spectrophotometer and Bioanalyzer 2100 system (Agilent, Santa Clara, CA, USA). RT-qPCRs were performed using a Rotor-Gene Q (Qiagen, Hilden, Germany). Each run was assayed in triplicate in a total volume of 20 µL containing the RNA template, One Step RT-PCR SYBR Mix (Doctor Protein, Korea), reverse transcriptase, and 10 pmole of each primer. Reverse transcription was carried out at 42 °C for 30 min. PCR conditions were 95 °C/5 min for one cycle, followed by 95 °C and 55 °C/30 s for 40 cycles. Amplification of one specific target DNA was checked by melting curve analysis. The expression ratios were normalized to reference gene *ef1α* expression [35,36] and calculated according to the ΔΔCq method [37]. The expression stability of *ef1α* was determined by BestKeeper index via RefFinder (https://www.heartcare.com.au/reffinder/, accessed on 2 April 2021) [38] and PCR efficiencies of studied genes were 89.5–101.8%. Expression of target genes mRNA was analyzed with appropriate oligonucleotide pairs (Appendix A). For RNA-seq analyses, 3-days-old culture of WT and mutant strains were harvested from solid MMY. Total RNA was extracted and submitted to eBiogen Inc. (Seoul, Korea) for library preparation and sequencing.

### 4.4. Measurement of Germination Rate and Yield of Conidia

To examine conidia germination levels, conidia of WT and mutants were inoculated into 5 mL MMY broth at a concentration of 2 × 10^5^ conidia/mL and incubated at 37 °C. Two hours after inoculation, germination was assessed every 2 h. Three random visual fields were observed microscopically. The percentage of germination was calculated by the number of total conidia and germinated conidia in the visual field. To determine conidia number, conidial suspension (about 10^6^ conidia) of three strain was spread on to MMY solid media and incubated in the dark at 37 °C. At the indicated time, conidia were collected with 0.5% Tween 80 solution from the plate and filtered through Miracloth (Calbiochem, San Diego, CA, USA), and counted using a hemocytometer. 

### 4.5. Determination of the Conidial Hydrophobicity

The hydrophobicity of conidia cell wall was determined as described previously with a detergent solution (0.2% SDS, 50 mM EDTA) [8]. Hydrophobicity was also assessed by aqueous-solvent partitioning assays, using the microbial adhesion to solvents (MATS) method [39]. Briefly, 2 mL of conidial suspension in 0.1 M KNO_3_ (2 to 7 × 10^6^ conidia/mL) were vortexed vigorously with 400 μL of hexadecane for 2 min. After separation of the two phases, an aliquot of the aqueous phase was collected, and the number of conidia in the aqueous phase was determined using a hemocytometer. The percentage of bound conidia to solvent was calculated as follows: % adhesion to solvent = (1 − N/N_0_) × 100 where N_0_ is the initial number of conidia in the aqueous phase and N is the residual number of conidia in the aqueous phase after partitioning. The RodA protein was extracted by incubating dry spores with hydrofluoric acid (HF) (10 μL per mg dry weight) for 72 h at 4 °C [25]. The full obtained protein was reconstituted in Laemmli’s sample buffer, and subjected to SDS-PAGE analysis and visualized by Coomassie Brilliant Blue staining.

### 4.6. Microscopy

TEM and AFM analysis was carried out at the Korea Basic Science Institute. For TEM analysis, conidia were fixed in 2.5% glutaraldehyde in 0.1 M phosphate, washed three times with 0.1 M phosphate, post-fixed in 1% osmium tetroxide, incubated for 1 h in 0.1 M phosphate, and dehydrated for 15 min in a graded methanol series from 50% to 100%. Samples were embedded in Epon resin 812. The sections were examined with a Tecnai G2 Spirit Twin Bio-Transmission Electron Microscope (FEI, Hillsboro, OR, USA), with an accelerating voltage of 120 KV. Conidial surfaces were analyzed by a Nanoscope V Multimode 8 AFM (Bruker, Santa Barbara, CA, USA). Conidia were immobilized by mechanically trapping them into porous polycarbonate membranes. After filtering a concentrated suspension of conidia, the filter was rinsed with deionized water, carefully cut, and attached to a metallic puck using double-sided sticky tape. Images were performed in soft tapping mode using a silicon AFM probe (k = 42 N/m, f = 320 kHz, NCHR, Nanoworld, Neuchâtel, Switzerland). 

### 4.7. Transcriptome Analysis

For control and test RNAs, the construction of library was performed using QuantSeq 3′ mRNA-Seq Library Prep Kit (Lexogen, Inc., Wien, Austria) according to the manufacturer’s instructions. High-throughput sequencing was performed as single-end 75 sequencing using NextSeq 500 (Illumina, Inc., San Diego, CA, USA). QuantSeq 3’ mRNA-Seq reads were aligned using Bowtie2 [40]. Bowtie2 indices were either generated from genome assembly sequence or the representative transcript sequences for aligning to the genome and transcriptome. The alignment file was used for assembling transcripts, estimating their abundances and detecting differential expression of genes. Differentially expressed gene were determined based on counts from unique and multiple alignments using coverage in Bedtools [41]. The RT (Read Count) data were processed based on Quantile normalization method using EdgeR within R (R development Core Team, 2016) using Bioconductor [42]. Gene classification was based on searches done by DAVID (http://david.abcc.ncifcrf.gov/, accessed on 15 July 2018) and Medline databases (http://www.ncbi.nlm.nih.gov/, accessed on 15 July 2018). Hierarchical clustering performed using ExDEGA (Excel based Differentially Expressed Gene Analysis) Program (ver. 3.0, ebiogen Inc., Seoul, Korea).

### 4.8. Detection of Gliotoxin (GT)

Amount of GT was determined by the thin layer chromatography (TLC) method as described previously [43]. Conidia (about 10^5^) of each strain were inoculated into 5 mL of liquid complete medium and cultured at 37 °C for 7 days under dark conditions. After incubation, an equal amount of chloroform was added per sample. Samples were centrifuged for 10 min. The separated organic phase was transferred to new glass vials and evaporated. Samples were resuspended in 50 μL of methanol and 10 μL loaded into a thin-layer chromatography (TLC). The TLC Silicagel 60 plate was developed with toluene/ethyl acetate/formic acid (5:4:1, *v*/*v*/*v*).

### 4.9. Murine Virulence Assay

For the immunocompromised mouse model, we used outbred CrlOri: CD1 (ICR) (Orient Bio Inc., Korea) female mice (30 g in body weight, 6 to 8 weeks old), which were housed five per cage and had access to food and water ad libitum. Mice were immunosuppressed by treatment of cyclophosphamide and cortisone (Figure 8A). For conidia inoculation, mice were anesthetized with isoflurane and then intranasally infected with 1 × 10^7^ conidia of *A*. *fumigatus* strains (10 mice per each fungal strain) in 30 µL of 0.01% Tween 80 in PBS. Mice were monitored every 12 h for survival for 5 days after the challenge. Mock mice included in all experiments were inoculated with sterile 0.01% Tween 80 in PBS. Mice were checked every 12 h for survival and Kaplan-Meier survival curves were analyzed using the Log-Rank (Mantel-Cox) test for significance (*p* < 0.05).

### 4.10. Immunoblotting

Total soluble proteins were extracted from WT, mutant, and C′ strains subjected to 0 and 20 min with 10 mg/mL CFW treatment. Samples were separated using 12% SDS-PAGE and blotted onto nitrocellulose membrane. Blot was analyzed with anti-phospho-p38 MAPK antibody (New England Biolabs, MA, USA) and anti-phospho-p44/42 MAPK antibody (Cell Signaling Technologies, Danvers, MA, USA). 

## Figures and Tables

**Figure 1 ijms-22-03777-f001:**
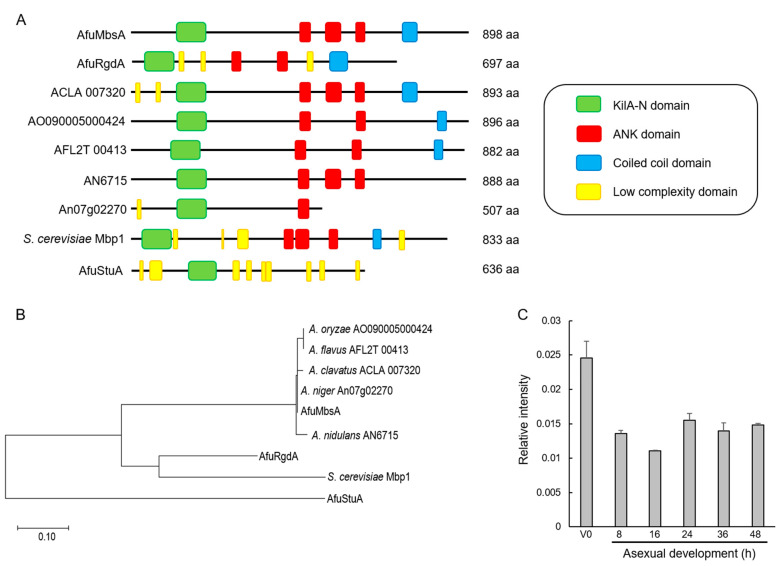
Summary of MbsA. (**A**) Schematic presentation of the domain structure of the MbsA-like proteins using SMART (http://smart.embl-heidelberg.de, accessed on 10 February 2021). (**B**) A phylogenetic tree of the MbsA-like proteins in various *Aspergillus* and *Saccharomyces cerevisiae* was constructed based on the matrix of pair-wise distances between the KilA-N domain sequences. (**C**) Levels of *mbsA* mRNA during the lifecycle of *A*. *fumigatus* wild type (WT, AF293). The vegetative stage (V0) and time (hours) of incubation in post asexual developmental induction is shown.

**Figure 2 ijms-22-03777-f002:**
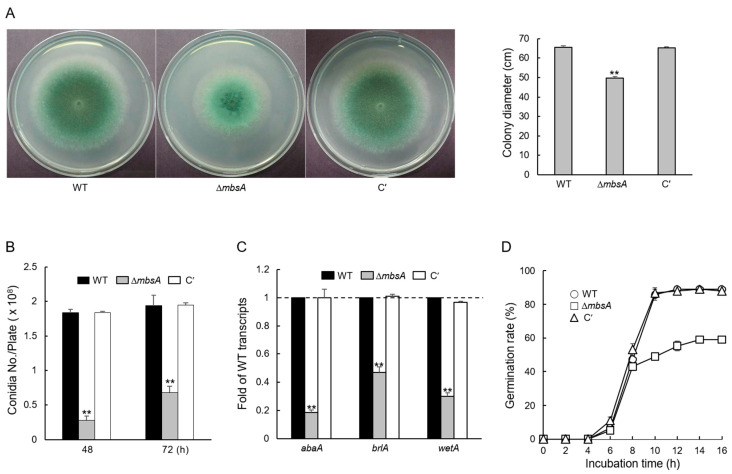
MbsA is required for proper growth, development, and spore germination. (**A**) Colony photographs and radial growth rate of wild type (WT), Δ*mbsA*, and complemented (C′) strains point-inoculated on solid glucose minimal medium with 0.1% yeast extract (MMY) and grown for 3 days. (**B**) Conidia numbers produced by each strain per plate. (**C**) mRNA levels of the key asexual developmental regulators in the Δ*mbsA* strain relative to WT at 3 days determined by RT-qPCR. Fungal cultures were done in solid MMY and mRNA levels were normalized using the *ef1α* gene. (**D**) Germination rates of *A*. *fumigatus* strains when inoculated in liquid CM at 37 °C. The number of conidia showing germ-tube protrusion was recorded at 2 h intervals and which is represented as a percentage of the total number of conidia in each microscope field. Data are presented as the mean ± standard deviation from three independent experiments. ANOVA test: ** *p* < 0.01.

**Figure 3 ijms-22-03777-f003:**
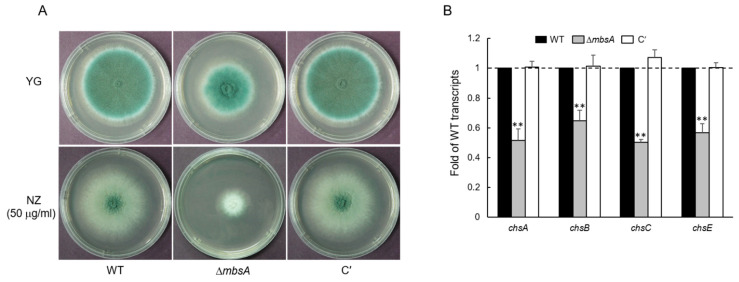
MbsA is necessary for the proper regulation of chitin synthesis. (**A**) Radial growth of WT, Δ*mbsA*, and C′ strains in the presence of Nikkomycin Z (NZ, 50 µg/mL) following incubation at 37 °C for 72 h. (**B**) Chitin synthesis–related genes in WT, Δ*mbsA*, and C′ strains analyzed by RT-qPCR. The *ef1α* gene as the endogenous control. Statistical differences between strains were evaluated with ANOVA test: ** *p* < 0.01.

**Figure 4 ijms-22-03777-f004:**
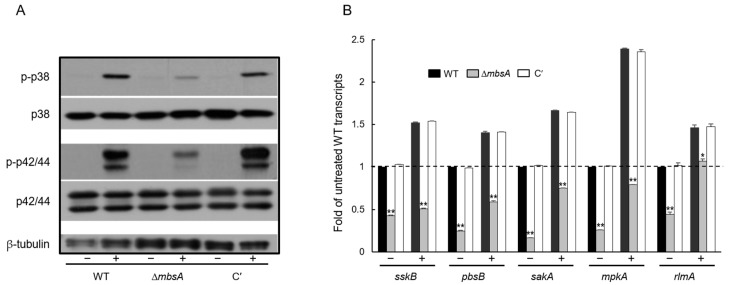
MbsA positively affects the SakA MAPK pathway. (**A**) Conidia of the indicated strains were grown for 14 h in MMY medium and treated with 200 µg/mL of Calcofluor White (CFW) (+) or not (−). Aliquots of cell were harvested after 20 min and used to prepare total protein extracts. Protein extracts were analyzed by immunoblotting with anti-phospho-p38 and anti-phoshpo-p42/44 antibodies. (**B**) mRNA levels of SakA MAP kinase pathway-related genes in WT, Δ*mbsA*, and C′ strains analyzed by RT-qPCR at the same culture condition. The *ef1α* gene as the endogenous control. Statistical differences between strains were evaluated with ANOVA test: ** *p* < 0.01, * *p* < 0.05.

**Figure 5 ijms-22-03777-f005:**
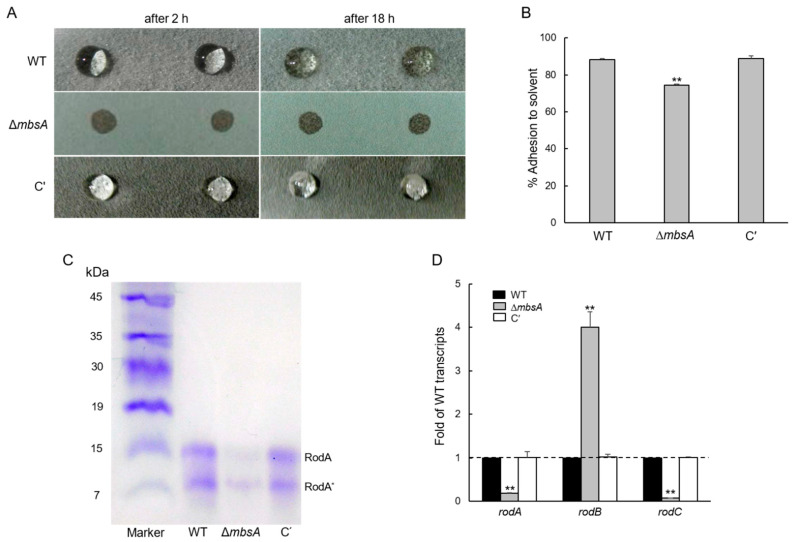
MbsA is needed for proper conidia hydrophobicity. (**A**) Hydrophobicity test for each strain. Strains were cultured on MMY agar plates for 4 days at 37 °C and 10 μL of a detergent solution (0.2% SDS, 50 mM EDTA) were dropped onto the surface of a colony. The droplets were observed to penetrate into the colonies. (**B**) Percentage hydrophobicity of three strains by MATS test. (**C**) SDS-PAGE analysis of the hydrophobin, RodA of relevant strains. RodA was extracted from the dried conidia (10^9^) with hydrofluoric acid (HF). RodA*: degraded form of RodA due to HF treatment. (**D**) RT-qPCR analysis of hydrophobin genes in WT, Δ*mbsA*, and C′ strains. The *ef1α* gene as the endogenous control. Statistical differences between strains were evaluated with ANOVA test: ** *p* < 0.01.

**Figure 6 ijms-22-03777-f006:**
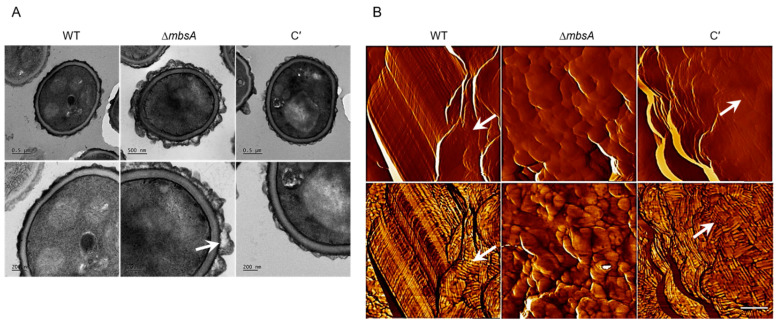
MbsA is needed for proper cell wall architecture and rodlet layer. (**A**) Transmission electron micrographs of conidia. The conidia were harvested from the cells cultured on MMY medium for 5 days, and they were fabricated as ultra-thin specimens for transmission electron microscopy. (**B**) Atomic force microscopy (AFM) image of conidial surface of three strains. Upper: amplitude image; Lower: phase image. Rodlets showed only in WT and C′ strains (arrows). Bar indicates 100 nm.

**Figure 7 ijms-22-03777-f007:**
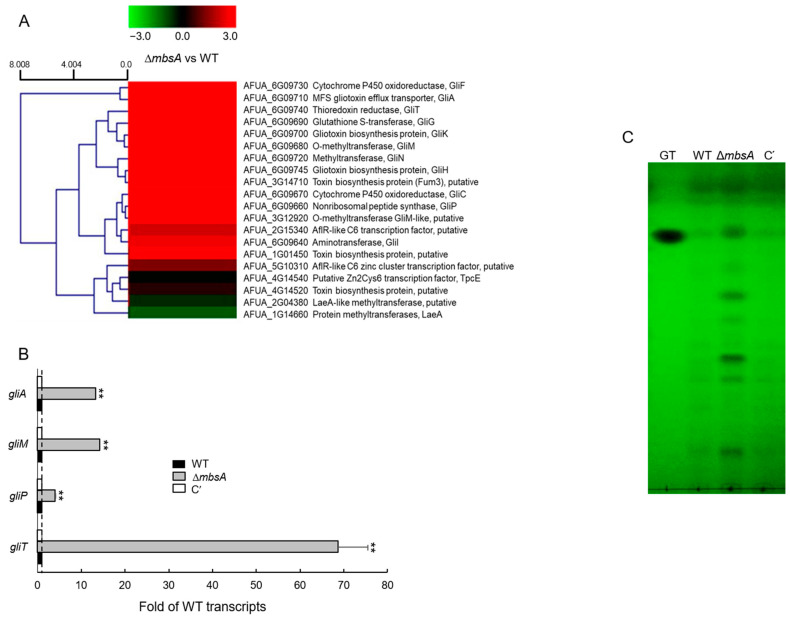
MbsA down-regulates gliotoxin (GT) production. (**A**) Heat map of those genes encoding toxin-related proteins. Most of gliotoxin biosynthetic genes were up-regulated by the loss of *mbsA*. (**B**) Determination of GT production in WT, Δ*mbsA*, and C′ strains. The culture supernatant of each strain was extracted with chloroform and subjected to TLC. (**C**) RT-qPCR analysis of GT-related genes in WT, Δ*mbsA*, and C′ strains. The *ef1α* gene as the endogenous control. Statistical differences between WT and mutant strains were evaluated with ANOVA test: ** *p* < 0.01.

**Figure 8 ijms-22-03777-f008:**
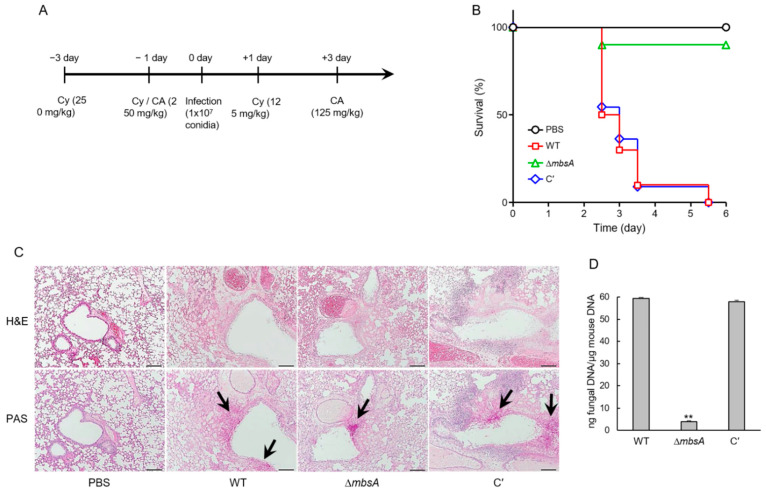
MbsA’s role in virulence. (**A**) Schematic presentation of *A*. *fumigatus* infection in immunocompromised mouse model. Six-week-old female ICR mice were immunocompromised by treatment of cyclophosphamide (Cy, 250 mg/kg at day −3 and −1 and 125 mg/kg at day +1) and cortisone acetate (CA, 250 mg/kg at day −1 and 125 mg/kg at day +3). On day 0 mice were intranasally infected. (**B**) Survival curve of mice infected with WT, Δ*mbsA*, and C′ strains (*n* = 10/group). (**C**) Lung sections after each strains infection were stained with Hematoxylin and Eosin (H&E) or Periodic-acid Schiff (PAS). Arrows indicate fungal mycelium. Bars indicate 200 µm. (**D**) Fungal burden in the lungs of mice infected with WT and Δ*mbsA*, and C′ strains. Data are represented as mean ± standard deviation from three independent experiments. ANOVA test: ** *p* < 0.01.

**Figure 9 ijms-22-03777-f009:**
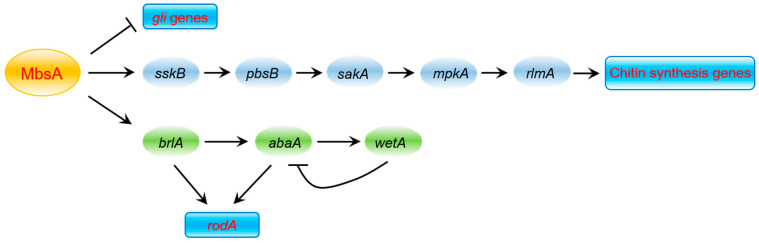
A genetic model depicting the role of MbsA in *A*. *fumigatus*. MbsA positively regulates asexual sporulation, *rodA* expression, the SakA MAP kinase pathway, and virulence, but negatively controls expression of gliotoxin-related genes and GT production.

## Data Availability

The RNA-seq data are available from NCBI Gene Expression Omnibus (GEO) database (GSE123744).

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
