# Peer review of "Characterization of the mbsA Gene Encoding a Putative APSES Transcription Factor in Aspergillus fumigatus"

_ijms, 2021, doi:10.3390/ijms22073777_

Round 1

Reviewer 1 Report

One of the fungal APSES family transcription factors was investigated for A. fumigatus. Many phenotypes of the gene deletion mutant were demonstrated, which will be useful information for the researchers in this field. However, there are many concerns on experimental methods, which are critical for appropriate interpretation of the function of the gene of interest. For considering acceptance for publication, they should be properly addressed in the revised manuscript.

Major concerns

  • The name of the gene (mbpA) of interest is very confusing, because the closer A. fumigatus protein to S. cerevisiae Mpb1 is RgdA. The authors should reconsider the name of protein that was characterized in this work. For this help, alignment of KilA-N domains of all the protein mentioned here and the phylogenetic tree for this also should be presented. This will highlight which A. fumigatus APSES proteins correspond to those of S. cerevisiae (Asm1p, Phd1p, Sok2p, Efg1p, and StuAp). I also wonder if the Swi6 is an APSES protein?
  • 2B: The conidia productivity apparently decreased in the mutant. However, the mutant showed delayed colony growth, which means progression of conidiation area is also slowly expanded. Therefore, total conidia numbers from the different sized colonies do not reflect “real” conidia productivity. The authors should harvest the conidia from a plate, where fungi were not point inoculated (like the plate pictured in Fig. 3B, but without a hole).
  • For gene expression analysis, the authors are looking at the gene related to conidiation or conidia property. Expression level in liquid culture provides very limited information. Therefore, the fungal cells should be cultured up to asexual development stage (Fig. 2C and Fig. 5D). In addition, to investigate the effect of CFW on expression of the genes related to SakA and Cell wall integrity pathway, the cells should be cultured upon CFW treatment. Which gene was used as an internal control? This should be present in the method section and each figure legend.
  • 2D: it seems that germination kinetics were very similar between WT and the mutant strain based on the rate of germinated conidia in 4 to 8 h. “Germination rates” at 8h were almost same among the strains. Meanwhile, around 60% of the mutant conidia is finally germinated, which means 40% were dead conidia. Therefore, there is a difference in “survival rate” of the conidia between WT and the mutant. The authors should be careful to use the terms “germination kinetics” and “germination rate”.
  • 3A: The colony size for WT was apparently affected by CFW and CR. However, the mutant colony retained the size regardless of cell wall stressor treatment. This seems to be contradicting the author’s view that MbpA is necessary for the protection against cell wall stresses. Please reconsider the interpretation or provide additional evidence that support your thought.
  • 3B: Only from the data, it is unclear if the mutant strains showed susceptible to Nikkomycin Z. Clear inhibition zone was not seen around the hole both in WT and the mutant. Please provide another data supporting the conclusions.
  • 4A: Western blot for the MAPKs need s total protein control for each kinase. For example, Anti-p38 should be used in parallel for that with Anti-p-p38. Without the information, it is unclear if total amount of the kinase protein was reduced or the rate of phosphorylated kinase was decreased. Please provide the data for total kinase protein in the blotting experiment.
  • 4B: the authors investigated if the MAPKs are phosphorylated upon CFW treatment. In this context, expression analysis should be conducted in the presence of CFW.
  • 7A: Heat map of expression of the genes related to toxin. I wondered what is the definition to select the toxin-related genes from about 10,000 genes. The method to do hierarchical clustering is also lacking.
  • In the model proposed by the authors in Fig. 9, I wonder if the chs genes are regulated by SakA and MpkA pathway? More explanations to support the model should be provided in the discussion section.

Minor concerns

  • What is PQ
  • Please provide the growth condition to extract GT.

Author Response

# Reviewer 1

One of the fungal APSES family transcription factors was investigated for A. fumigatus. Many phenotypes of the gene deletion mutant were demonstrated, which will be useful information for the researchers in this field. However, there are many concerns on experimental methods, which are critical for appropriate interpretation of the function of the gene of interest. For considering acceptance for publication, they should be properly addressed in the revised manuscript.

Major concerns

The name of the gene (mbpA) of interest is very confusing, because the closer A. fumigatus protein to S. cerevisiae Mpb1 is RgdA. The authors should reconsider the name of protein that was characterized in this work. For this help, alignment of KilA-N domains of all the protein mentioned here and the phylogenetic tree for this also should be presented. This will highlight which A. fumigatus APSES proteins correspond to those of S. cerevisiae (Asm1p, Phd1p, Sok2p, Efg1p, and StuAp). I also wonder if the Swi6 is an APSES protein?

⇒ We appreciate this valuable comment. We used the Afu7g05620 gene name according to NCBI database and domain structure and phylogenetic tree reconstruct as your suggestion (Fig. 1).

2B: The conidia productivity apparently decreased in the mutant. However, the mutant showed delayed colony growth, which means progression of conidiation area is also slowly expanded. Therefore, total conidia numbers from the different sized colonies do not reflect “real” conidia productivity. The authors should harvest the conidia from a plate, where fungi were not point inoculated (like the plate pictured in Fig. 3B, but without a hole).

⇒ Thank you for your comment! We entirely agree your comments and we determined conidia number as “conidia number per growth area”. So, we revised Fig. 2B and Line 87-88.

For gene expression analysis, the authors are looking at the gene related to conidiation or conidia property. Expression level in liquid culture provides very limited information. Therefore, the fungal cells should be cultured up to asexual development stage (Fig. 2C and Fig. 5D).

⇒ We are very sorry to make such a mistake. We analyzed gene expression levels with solid MMY medium-cultured samples, and described it (Line 103).

In addition, to investigate the effect of CFW on expression of the genes related to SakA and Cell wall integrity pathway, the cells should be cultured upon CFW treatment. Which gene was used as an internal control? This should be present in the method section and each figure legend.

⇒ Thank you again! We determined the expression of SakA MAP kinase pathway genes in the presence of CFW (Line 142). The expression levels of target genes were normalized with the ef1α gene as the internal control and described in the method section (Line 346-348) and each figure legend.

2D: it seems that germination kinetics were very similar between WT and the mutant strain based on the rate of germinated conidia in 4 to 8 h. “Germination rates” at 8h were almost same among the strains. Meanwhile, around 60% of the mutant conidia is finally germinated, which means 40% were dead conidia. Therefore, there is a difference in “survival rate” of the conidia between WT and the mutant. The authors should be careful to use the terms “germination kinetics” and “germination rate”.

⇒ We appreciate your valuable comment. We corrected as “germination rate” (Line 92 and 103).

3A: The colony size for WT was apparently affected by CFW and CR. However, the mutant colony retained the size regardless of cell wall stressor treatment. This seems to be contradicting the author’s view that MbpA is necessary for the protection against cell wall stresses. Please reconsider the interpretation or provide additional evidence that support your thought.

⇒ Thank you so much for your significant comments. We changed the interpretation as “MbpA is necessary for the proper regulation of responses to cell wall stress” and revised it accordingly (Line 118).

3B: Only from the data, it is unclear if the mutant strains showed susceptible to Nikkomycin Z. Clear inhibition zone was not seen around the hole both in WT and the mutant. Please provide another data supporting the conclusions.

⇒ We appreciate again! We replaced the old one with new data (Fig. 3B).

4A: Western blot for the MAPKs need s total protein control for each kinase. For example, Anti-p38 should be used in parallel for that with Anti-p-p38. Without the information, it is unclear if total amount of the kinase protein was reduced or the rate of phosphorylated kinase was decreased. Please provide the data for total kinase protein in the blotting experiment.

⇒ Thank you! We provided total kinase blotting data accordingly (Fig. 4A).    

4B: the authors investigated if the MAPKs are phosphorylated upon CFW treatment. In this context, expression analysis should be conducted in the presence of CFW.

⇒ Thank you again! We determined the expression of SakA MAP kinase pathway genes in the presence of CFW and described the results (Line 142).

7A: Heat map of expression of the genes related to toxin. I wondered what is the definition to select the toxin-related genes from about 10,000 genes. The method to do hierarchical clustering is also lacking.

⇒ We appreciate your valuable comment. We selected genes by “Gene Ontology and GO Annotations” and hierarchical clustering performed using ExDEGA (Excel based Differentially Expressed Gene Analysis) Program (ver. 3.0, ebiogen Inc. Korea). The method was inserted in “Materials and methods” section (Line 401-403).

In the model proposed by the authors in Fig. 9, I wonder if the chs genes are regulated by SakA and MpkA pathway? More explanations to support the model should be provided in the discussion section.

⇒ Thank you again! We revised Fig. 9 and provided more explanations in the Discussion section (Line 263-264).

Minor concerns

What is PQ

⇒ Thanks! We revised as paraquat (PQ) (Line 250).

Please provide the growth condition to extract GT.

⇒ Thank you! We provided the growth condition in the Materials and methods section (Line 407-413).

Reviewer 2 Report

The authors are presenting the characterization of the putative  APSES TF mbpA in Aspergillus fumigatus. The paper is already in a good written form, except minor aspects (see below). What cut down my enthusiasm while reading is the fact that an almost identical paper (in the format and style, not the content) has been published already, by the same group of authors. Despite it is in general considered normal to maintain a very similar style while writing about a similar subject, the amount of self-plagiarism present in this paper is almost unbelievable.

The introduction appears like a badly summarized version of the one previously published, with the difference that the cut parts were quite necessary to fully understand the background.

The results part is almost a copy-paste work, just with a different gene name, and sometimes "increase" instead of "decrease". Of course if the experiments performed were the same one cannot invent a complete new way of describing them, but the extent of copy-work is striking. For example:

-(from the previous publication) 

RgdA affects mycelial growth and asexual sporulation.

To investigate functions of rgdA, we generated the ΔrgdA null mutant and complemented strains. The ΔrgdA mutant showed significantly reduced radial growth compared to wild-type (WT) and complemented strains (Fig. 2A). Quantitative analyses of numbers of conidia per plate grown on solid medium further demonstrated that asexual spore production in the ΔrgdA mutant (0.98 × 107 conidia/cm2) was dramatically decreased to about 35% of that in WT and complemented strains (Fig. 2B). In accordance with this observation, in the ΔrgdA mutant, mRNA levels of the key asexual developmental regulators abaA, brlA, and wetA were significantly reduced (Fig. 2C). These results suggest that RgdA is necessary for both normal growth and proper conidiation.

-(from this paper)

2.2. MbpA affects growth, asexual sporulation, and spore germination

To investigate functions of mbpA, we generated the mbpA null (Δ) mutant and complemented strains (C'). The ΔmbpA mutant showed about 25% reduced radial growth compared to wild type (WT) and C' strains (Figure 2A). Quantitative analyses of conidia per plate of the culture grown on solid medium further demonstrated that asexual spore production in the ΔmbpA mutant (1.48 × 107 conidia/cm2) was dramatically decreased to about 50% of that of WT and C' strains (Figure 2B). In accordance with this observation, in the ΔmbpA mutant, mRNA levels of key asexual developmental regulators, abaA, brlA, and wetA were significantly lowered (Figure 2C). These results suggest that MbpA is necessary for both proper growth and conidiation.

Highlighted are the part identical between the two papers. Every paragraph in the result part is the same.

The discussion part is the most disappointing. After referring so much to the previous publication, one would expect a constructive comparison between the effect of the two ASPES TF, RgdA and MbpA. At the end, they were characterized with the same assays and in some case they presented opposite results. A discussion covering up those similarities and differences, perhaps with an hypothesis about the reason, seems needed. But there is no mention of RgdA, only of StuA (and that part was already present in the previous paper).

Finally, the materials and methods, considering they have been basically described in the previous paper, they can be drastically reduced to a reference.

Despite not presenting any particular gap from the experimental point of view, the writing and the wrapping up of the results in the discussion must be improved.

Minor comments:

-in the introduction there is no background about the APSES TF family as well as nothing about Aspergillus fumigatus. However, there is the mention of Swi4-Swi6 from Saccharomyces Cerevisiae, with no clear connection to the context. That part was already covered in the previous paper about RgdA, and here is just popping out with no logic.

-in the RNA-Seq analyses, the wt and deletion strains were analysed. Then why there is a mention of significantly higher transcripts in the deletion strain  compared to WT and complemented strains (line 205)? Was the complemented strain checked as well?

- line 154-155 belongs more to the introduction than to the results   

Overall, the work behind this paper is clear, as it is the fact that MpbA has been newly characterized and presents some very interesting phenotype. The fact that quite often it is a common choice to divide one bigger paper into two to publish back to back is understandable; but I would recommend the authors to try and change a bit more the format to make it looks a bit less  like a copy and paste job. And when the second part is written, it would be more useful to openly refer to the first , already published part of the work, and create comparison between the two characterizations.

Author Response

# Reviewer 2

The authors are presenting the characterization of the putative APSES TF mbpA in Aspergillus fumigatus. The paper is already in a good written form, except minor aspects (see below). What cut down my enthusiasm while reading is the fact that an almost identical paper (in the format and style, not the content) has been published already, by the same group of authors. Despite it is in general considered normal to maintain a very similar style while writing about a similar subject, the amount of self-plagiarism present in this paper is almost unbelievable.

The introduction appears like a badly summarized version of the one previously published, with the difference that the cut parts were quite necessary to fully understand the background.

⇒ We appreciate your valuable comment. We revised “Introduction” section as suggested.

The results part is almost a copy-paste work, just with a different gene name, and sometimes "increase" instead of "decrease". Of course if the experiments performed were the same one cannot invent a complete new way of describing them, but the extent of copy-work is striking. For example:

-(from the previous publication)

RgdA affects mycelial growth and asexual sporulation.

To investigate functions of rgdA, we generated the ΔrgdA null mutant and complemented strains. The ΔrgdA mutant showed significantly reduced radial growth compared to wild-type (WT) and complemented strains (Fig. 2A). Quantitative analyses of numbers of conidia per plate grown on solid medium further demonstrated that asexual spore production in the ΔrgdA mutant (0.98 × 107 conidia/cm2) was dramatically decreased to about 35% of that in WT and complemented strains (Fig. 2B). In accordance with this observation, in the ΔrgdA mutant, mRNA levels of the key asexual developmental regulators abaA, brlA, and wetA were significantly reduced (Fig. 2C). These results suggest that RgdA is necessary for both normal growth and proper conidiation.

-(from this paper)

2.2. MbpA affects growth, asexual sporulation, and spore germination

To investigate functions of mbpA, we generated the mbpA null (Δ) mutant and complemented strains (C'). The ΔmbpA mutant showed about 25% reduced radial growth compared to wild type (WT) and C' strains (Figure 2A). Quantitative analyses of conidia per plate of the culture grown on solid medium further demonstrated that asexual spore production in the ΔmbpA mutant (1.48 × 107 conidia/cm2) was dramatically decreased to about 50% of that of WT and C' strains (Figure 2B). In accordance with this observation, in the ΔmbpA mutant, mRNA levels of key asexual developmental regulators, abaA, brlA, and wetA were significantly lowered (Figure 2C). These results suggest that MbpA is necessary for both proper growth and conidiation.

Highlighted are the part identical between the two papers. Every paragraph in the result part is the same.

⇒ We appreciate your valuable comment. We revised every paragraph accordingly (Line 83-93).

The discussion part is the most disappointing. After referring so much to the previous publication, one would expect a constructive comparison between the effect of the two ASPES TF, RgdA and MbpA. At the end, they were characterized with the same assays and in some case they presented opposite results. A discussion covering up those similarities and differences, perhaps with an hypothesis about the reason, seems needed. But there is no mention of RgdA, only of StuA (and that part was already present in the previous paper).

⇒ We truly appreciate this constructive comment. We have done our best to improve Discussion by integrating RgdA and StuA information with additional discussion (Line 244-245, 248, 268, 281, 295-298).

Finally, the materials and methods, considering they have been basically described in the previous paper, they can be drastically reduced to a reference.

⇒ Thanks again! While we in part agree with this reviewer’s comment, per request from the editor, we added additional description (Line 334-348, 401-403, 407-413).

Despite not presenting any particular gap from the experimental point of view, the writing and the wrapping up of the results in the discussion must be improved.

⇒ Thank you for your valuable comments! We tried to improve Discussion section accordingly (Line 244-245, 248, 268, 281, 295-298).

Minor comments:

-in the introduction there is no background about the APSES TF family as well as nothing about Aspergillus fumigatus. However, there is the mention of Swi4-Swi6 from Saccharomyces Cerevisiae, with no clear connection to the context. That part was already covered in the previous paper about RgdA, and here is just popping out with no logic.

⇒ We appreciate this valuable comment. We added background about the APSES TF family revised as your suggestions (Line 42-44, Line 52-54).

-in the RNA-Seq analyses, the wt and deletion strains were analysed. Then why there is a mention of significantly higher transcripts in the deletion strain compared to WT and complemented strains (line 205)? Was the complemented strain checked as well?

⇒ Thank you for your kind comments! We added description about significant DEGs (Line 195-196) and the expression levels of these genes were checked complemented strain as well by qRT-PCR. But, due to limited resources, we were not able to perform RNA-seq analysis using a complemented strain.

- line 154-155 belongs more to the introduction than to the results  

⇒ Thank you! We deleted line 154-155.

Overall, the work behind this paper is clear, as it is the fact that MpbA has been newly characterized and presents some very interesting phenotype. The fact that quite often it is a common choice to divide one bigger paper into two to publish back to back is understandable; but I would recommend the authors to try and change a bit more the format to make it looks a bit less like a copy and paste job. And when the second part is written, it would be more useful to openly refer to the first, already published part of the work, and create comparison between the two characterizations.

⇒ We appreciate this valuable comment. We tried to revise as your suggestions (Discussion section, Line 83 and 144).

Round 2

Reviewer 1 Report

The manuscript was in part adequately changed, however there are still much to be improved. I am afraid that my previous comments were properly understood to the authors. Again, I would like to ask same concerns, which require some additional experiments and adequate answers supporting the author’s claims.

  • The name of the gene (mbpA) of interest is very confusing, because the closer A. fumigatus protein to S. cerevisiae Mpb1 is RgdA. The authors should reconsider the name of protein that was characterized in this work. For this help, alignment of KilA-N domains of all the protein mentioned here and the phylogenetic tree for this also should be presented. This will highlight which A. fumigatus APSES proteins correspond to those of S. cerevisiae (Asm1p, Phd1p, Sok2p, Efg1p, and StuAp). I also wonder if the Swi6 is an APSES protein?

⇒ We appreciate this valuable comment. We used the Afu7g05620 gene name according to NCBI database and domain structure and phylogenetic tree reconstruct as your suggestion (Fig. 1).

>My comment was not adequately understood. The protein (Afu7g05620) is not an ortholog of S. cerevisiae Mbp1. The sentence in L71-73 is totally wrong. 25.7% of identity is too low. Therefore, I think the A. fumigatus protein should not be named after Mbp1 and can be renamed. If the phylogenetic tree was constructed using KilA-N domains, this should be noted in the figure legend. Again, I ask if the Swi6 is an APSES protein?

  • 2B: The conidia productivity apparently decreased in the mutant. However, the mutant showed delayed colony growth, which means progression of conidiation area is also slowly expanded. Therefore, total conidia numbers from the different sized colonies do not reflect “real” conidia productivity. The authors should harvest the conidia from a plate, where fungi were not point inoculated (like the plate pictured in Fig. 3B, but without a hole).

⇒ Thank you for your comment! We entirely agree your comments and we determined conidia number as “conidia number per growth area”. So, we revised Fig. 2B and Line 87-88.

>I believe that vegetative growth was delayed in the mutant strain. However, I think the conidia productivity cannot be properly examined by counting conidia number from point-inoculated colonies on agar. Because the colony size differs between the strains, it is difficult to compare the conidia productivity at same growth stage. The conidia should be harvested from the plate where the strains were grown on whole plate, not by point-inoculated colony. This would prevent from uneven growth area due to delayed vegetative growth. Time-course sampling can provide more reliable comparison.

  • For gene expression analysis, the authors are looking at the gene related to conidiation or conidia property. Expression level in liquid culture provides very limited information. Therefore, the fungal cells should be cultured up to asexual development stage (Fig. 2C and Fig. 5D).

⇒ We are very sorry to make such a mistake. We analyzed gene expression levels with solid MMY medium-cultured samples, and described it (Line 103).

>If you harvested the mycelia from 1d colony on solid MMY plate, please describe precisely in the Material and Method section how many conidia you inoculated, and how inoculated the conidia (by point inoculation?) and collected the mycelia from the plate. I could not find the information from the ref. 28. In any event, how do you think growth delay in the mutant affected initiation of asexual development? This should be taken into consideration.

Another concern about culture condition is for Figure 1C. The authors did not provide any information on culture condition in the legend and M&M section. Please provide the detailed information for it, not only by showing reference.

  • In addition, to investigate the effect of CFW on expression of the genes related to SakA and Cell wall integrity pathway, the cells should be cultured upon CFW treatment. Which gene was used as an internal control? This should be present in the method section and each figure legend.

⇒ Thank you again! We determined the expression of SakA MAP kinase pathway genes in the presence of CFW (Line 142). The expression levels of target genes were normalized with the ef1α gene as the internal control and described in the method section (Line 346-348) and each figure legend.

>The authors showed phosphorylation of SakA and MpkA upon CFW treatment compared with that without CFW treatment. Accordingly, the data for transcript level should be shown in the same manner. This would contribute to discussing how the MbpA protein is involved in activation in the HOG pathway.

  • 3A: The colony size for WT was apparently affected by CFW and CR. However, the mutant colony retained the size regardless of cell wall stressor treatment. This seems to be contradicting the author’s view that MbpA is necessary for the protection against cell wall stresses. Please reconsider the interpretation or provide additional evidence that support your thought.

⇒ Thank you so much for your significant comments. We changed the interpretation as “MbpA is necessary for the proper regulation of responses to cell wall stress” and revised it accordingly (Line 118).

>The question which I raised was not adequately answered. What is “proper” regulation? It is absolutely obscure. My question is whether the MbpA function in cell wall stress resistance in a positive manner? To clarify this, it would be preferred to point-inoculate each strain on plates with and without CFW and CR (and Nikkomycin Z?) in different concentrations. Please use different plate for each strain to prevent from colonies of the strains affecting each other.

  • 3B: Only from the data, it is unclear if the mutant strains showed susceptible to Nikkomycin Z. Clear inhibition zone was not seen around the hole both in WT and the mutant. Please provide another data supporting the conclusions.

⇒ We appreciate again! We replaced the old one with new data (Fig. 3B).

>From the data, I just think the mutant was inhibited in conidiation by Nikkomycin Z. How about showing the colony growth data instead?

Author Response

I upload our response.

Please check upload file.

Round 3

Reviewer 1 Report

>My comment was not adequately understood. The protein (Afu7g05620) is not an ortholog of S. cerevisiae Mbp1. The sentence in L71-73 is totally wrong. 25.7% of identity is too low. Therefore, I think the A. fumigatus protein should not be named after Mbp1 and can be renamed. If the phylogenetic tree was constructed using KilA-N domains, this should be noted in the figure legend. Again, I ask if the Swi6 is an APSES protein?

⇒We appreciate your valuable comments. We deleted L71-73 and revised the sentences as shown in Line 73-77. In addition, we agree with the gene nomenclature issue, and renamed the protein as MbsB (Line 59-61) throughout the manuscript.

>If you named after Cryptococcus neoformans Mbs1, it can be MbsA. Why did you use MbsB? Alternatively, as shown in the figure that you provided in the letter, the protein belongs to the clade A-II together with A. nidulans Swi6 (AN6715). Therefore, Swi6 is also acceptable.

Anyway, this protein is shown as AfuMbsA in the Fig. 1A and 1B. Please correct properly.

L241-244: MbsB of A. fumigatus is phylogenetically related with MbsB-like proteins of other Aspergillus, but not with RgdA of the same fungus, or Mbp1 and Swi6 of budding yeast S. cerevisiae (Figure 1). This sentence seems strange, in particular the first part makes no sense. These proteins are likely to be orthologs not MbsB-like. In Fig. 1B, there is not Swi6. Please include it.

>If you harvested the mycelia from 1d colony on solid MMY plate, please describe precisely in the Material and Method section how many conidia you inoculated, and how inoculated the conidia (by point inoculation?) and collected the mycelia from the plate. I could not find the information from the ref. 28. In any event, how do you think growth delay in the mutant affected initiation of asexual development? This should be taken into consideration.

⇒Thank you! We deleted ref. 28 and described the detailed methods in Line 321-324. We also discussed a possible reason for delayed growth in Line 246-250 and Supplementary Fig. S1

>Fig. 2C are showing the gene expressions at 24h on solid MMY, which are conflict with the added description in the Material Method. Please provide the data for culturing for 3d instead.

>The question which I raised was not adequately answered. What is “proper” regulation? It is absolutely obscure. My question is whether the MbpA function in cell wall stress resistance in a positive manner? To clarify this, it would be preferred to point-inoculate each strain on plates with and without CFW and CR (and Nikkomycin Z?) in different concentrations. Please use different plate for each strain to prevent from colonies of the strains affecting each other.>From the data, I just think the mutant was inhibited in conidiation by Nikkomycin Z. How about showing the colony growth data instead?

⇒Thank you again! We performed experiments as your suggestions. However, there were no significant growth differences with the treatment of CR and CFW. The growth of mutant strains was inhibited only in the treatment of Nikkomycin Z. So, we focused on the chitin synthesis and revised Fig. 3, Fig. 9,Line 122-123 and 254-255, accordingly

>Fig. 3B can be removed.

Author Response

Thank you for your recomendations.

Comments and Suggestions for Authors

>My comment was not adequately understood. The protein (Afu7g05620) is not an ortholog of S. cerevisiae Mbp1. The sentence in L71-73 is totally wrong. 25.7% of identity is too low. Therefore, I think the A. fumigatus protein should not be named after Mbp1 and can be renamed. If the phylogenetic tree was constructed using KilA-N domains, this should be noted in the figure legend. Again, I ask if the Swi6 is an APSES protein?

⇒We appreciate your valuable comments. We deleted L71-73 and revised the sentences as shown in Line 73-77. In addition, we agree with the gene nomenclature issue, and renamed the protein as MbsB (Line 59-61) throughout the manuscript.

>If you named after Cryptococcus neoformans Mbs1, it can be MbsA. Why did you use MbsB? Alternatively, as shown in the figure that you provided in the letter, the protein belongs to the clade A-II together with A. nidulans Swi6 (AN6715). Therefore, Swi6 is also acceptable.

⇒ We truly appreciate this critical suggestion. Following your recommendation, we have named the gene/protein as MbsA and edited the entire manuscript accordingly. THANK YOU!

Anyway, this protein is shown as AfuMbsA in the Fig. 1A and 1B. Please correct properly.

⇒ We appreciate you finding this mistake. We have corrected it as MbsA throughout the manuscript.  THANK YOU again!  

L241-244: MbsB of A. fumigatus is phylogenetically related with MbsB-like proteins of other Aspergillus, but not with RgdA of the same fungus, or Mbp1 and Swi6 of budding yeast S. cerevisiae (Figure 1). This sentence seems strange, in particular the first part makes no sense. These proteins are likely to be orthologs not MbsB-like. In Fig. 1B, there is not Swi6. Please include it.

⇒ We truly appreciate your valuable insights and suggestions. Indeed, we checked the domain structure of the S. cerevisiae Swi6 protein and found that Swi6p lacks the KilA-N domain different from that of Aspergillus sp., Neurospora crassa, and many dimorphic fungi (Histoplasma, Coccidioides, etc). As a consequence, we could not include it in Fig. 1B, but have revised the sentence as follows (Line 240-242);

“MbsA of A. fumigatus is phylogenetically unrelated with RgdA of the same fungus, or Mbp1 of budding yeast S. cerevisiae (Figure 1)”.

>If you harvested the mycelia from 1d colony on solid MMY plate, please describe precisely in the Material and Method section how many conidia you inoculated, and how inoculated the conidia (by point inoculation?) and collected the mycelia from the plate. I could not find the information from the ref. 28. In any event, how do you think growth delay in the mutant affected initiation of asexual development? This should be taken into consideration.

⇒Thank you! We deleted ref. 28 and described the detailed methods in Line 321-324. We also discussed a possible reason for delayed growth in Line 246-250 and Supplementary Fig. S1

>Fig. 2C are showing the gene expressions at 24h on solid MMY, which are conflict with the added description in the Material Method. Please provide the data for culturing for 3d instead.

⇒ We thank you very much for finding this mistake. Despite several proof-reading and editing, somehow this mistake occurred. Accordingly, we changed 24 h to 3 days (Line 108). THANK YOU!

>The question which I raised was not adequately answered. What is “proper” regulation? It is absolutely obscure. My question is whether the MbpA function in cell wall stress resistance in a positive manner? To clarify this, it would be preferred to point-inoculate each strain on plates with and without CFW and CR (and Nikkomycin Z?) in different concentrations. Please use different plate for each strain to prevent from colonies of the strains affecting each other.>From the data, I just think the mutant was inhibited in conidiation by Nikkomycin Z. How about showing the colony growth data instead?

⇒Thank you again! We performed experiments as your suggestions. However, there were no significant growth differences with the treatment of CR and CFW. The growth of mutant strains was inhibited only in the treatment of Nikkomycin Z. So, we focused on the chitin synthesis and revised Fig. 3, Fig. 9,Line 122-123 and 254-255, accordingly

>Fig. 3B can be removed.

⇒ Thank you again! We removed Figure 3B following your suggestion (Line 119 and 123, Figure 3).
